# Studying Venom Toxin Variation Using Accurate Masses from Liquid Chromatography–Mass Spectrometry Coupled with Bioinformatic Tools

**DOI:** 10.3390/toxins16040181

**Published:** 2024-04-07

**Authors:** Luis L. Alonso, Jory van Thiel, Julien Slagboom, Nathan Dunstan, Cassandra M. Modahl, Timothy N. W. Jackson, Saer Samanipour, Jeroen Kool

**Affiliations:** 1Division of Bioanalytical Chemistry, Department of Chemistry and Pharmaceutical Sciences, Faculty of Sciences, Amsterdam Institute of Molecular and Life Sciences (AIMMS), Vrije Universiteit Amsterdam, 1081 HV Amsterdam, The Netherlands; l.lago.alonso@vu.nl (L.L.A.); j.slagboom@vu.nl (J.S.); 2Centre for Analytical Sciences Amsterdam (CASA), 1012 WX Amsterdam, The Netherlands; 3Institute of Biology Leiden, Leiden University, 2333 BE Leiden, The Netherlands; 4Naturalis Biodiversity Center, 2333 CR Leiden, The Netherlands; 5Venom Supplies Pty. Ltd., Tanunda, SA 5352, Australia; nathan@venomsupplies.com; 6Centre for Snakebite Research & Interventions, Liverpool School of Tropical Medicine, Liverpool L3 5QA, UK; cassandra.modahl@lstmed.ac.uk; 7Australian Venom Research Unit, Department of Biochemistry and Pharmacology, University of Melbourne, Parkville, VIC 3010, Australia; timothy.jackson@unimelb.edu.au; 8Van‘t Hof Institute for Molecular Sciences, University of Amsterdam, Science Park 904, 1098 XH Amsterdam, The Netherlands; s.samanipour@uva.nl

**Keywords:** LC-MS, snake venom, high throughput, data analysis

## Abstract

This study provides a new methodology for the rapid analysis of numerous venom samples in an automated fashion. Here, we use LC-MS (Liquid Chromatography–Mass Spectrometry) for venom separation and toxin analysis at the accurate mass level combined with new in-house written bioinformatic scripts to obtain high-throughput results. This analytical methodology was validated using 31 venoms from all members of a monophyletic clade of Australian elapids: brown snakes (*Pseudonaja* spp.) and taipans (*Oxyuranus* spp.). In a previous study, we revealed extensive venom variation within this clade, but the data was manually processed and MS peaks were integrated into a time-consuming and labour-intensive approach. By comparing the manual approach to our new automated approach, we now present a faster and more efficient pipeline for analysing venom variation. Pooled venom separations with post-column toxin fractionations were performed for subsequent high-throughput venomics to obtain toxin IDs correlating to accurate masses for all fractionated toxins. This workflow adds another dimension to the field of venom analysis by providing opportunities to rapidly perform in-depth studies on venom variation. Our pipeline opens new possibilities for studying animal venoms as evolutionary model systems and investigating venom variation to aid in the development of better antivenoms.

## 1. Introduction

Snake venoms are complex mixtures of proteins, peptides, and small molecules. These mixtures are produced in specialised oral glands and are transferred via the venom duct to fangs; glands, ducts, and fangs collectively form the snake venom delivery system [1]. Using this delivery system, venom toxins are injected into prey to facilitate their capture. These venom toxins immobilise prey by disrupting key pathophysiological pathways involving cardiovascular or nervous systems [2]. Therefore, the primary function of snake venom is prey acquisition [3,4,5]. Interestingly, there is no strong evidence suggesting that snake venom has evolved in order to be used as a defense mechanism [6]. The only exception can be seen in spitting cobras that evolved defensive venom-spitting behaviour to deter aggressors [7]. As predator and prey inevitably exert selection pressures upon each other, venom toxins—as molecules that mediate this antagonistic ecology—are at the centre of this co-evolutionary dynamic interplay. On one hand, prey evolve resistance to protect themselves [8]. On the other, snake venoms must maintain a selective advantage to overcome this resistance. Over millions of years, these reciprocal ecological interactions may have driven the evolution of the dramatic variation seen in snake venoms today [9].

Venom variation in snakes can be observed on the interspecific level (between species) and intraspecific level (within a species) [9]. While venom variation between species has been well studied (e.g., [4,5,10,11]), the extent of intraspecific variation is starting to gain significant attention [12]. A thorough understanding of venom variation is of interest to different scientific disciplines, as venom variation provides a model system for understanding adaptive responses to biotic and abiotic factors in an ecological context [5,7,13,14]. Additionally, snakebite envenoming is a neglected tropical disease severely affecting millions of people worldwide [15], and venom variation between and within species can complicate the cross-neutralising capacity of snakebite treatments [16]. For example, Russell’s viper (*Daboia russelii*) venoms of Bangladeshi populations are poorly neutralised by antivenoms that are solely manufactured using venoms from Indian snake populations [17]. As a result, the standard treatment dose must be drastically increased (i.e., five to ten times) to achieve effective neutralising of venom toxins in the body of a snakebite victim [17]. Furthermore, the diversity of bioactive venom toxins also provides a unique and promising resource for medicine, diagnostics, and cosmetics [18].

There is a need for analytical methods to characterise snake venoms in a high-throughput fashion to study their compositional diversity. However, there are limitations in the high-throughput analysis of large numbers of venoms. An important limitation in this regard when analysing snake venoms using Liquid Chromatography–Mass Spectrometry (LC-MS) is processing the LC-MS data acquired by integrating the peaks of all toxins found in each venom. This process will eventually result in a coherent list that can be used to look at venom variation at the toxin level. Due to the extensive number of proteins found in each venom [19] from gene duplication, isoform differentiation is paramount [20], and this is an arduous and tedious process that can also be tainted by manual errors both at the quantification of the peak areas—also known as integration—and data processing stages. These isoforms can result from post-translational modifications (PTMs), which change the mass of the protein toxin (compared to its amino acid sequence alone mass calculated by for example Mascot), and, depending on the specific protein, can also change the bioactivity of the toxins. This in turn has the consequence that eventual pathologies caused by venom toxins can also be mediated by PTMs on certain toxins [21].

To enable efficient venom variation studies based on LC-MS data, we established an automated analytical pipeline for the rapid characterisation and abundance comparison of toxins in large numbers of venoms. In our workflow, first venom toxins are separated using LC followed by MS detection with optional parallel nanofractionation via a post-column (1:9) flow splitter. After the flow split, the smaller fraction is sent to MS, and the larger fraction is transferred to a nanofraction collector for collecting high-resolution fractions in 384-well plates, which can then be used for high-throughput (HT) venomics characterisation [22] of the venom composition. Accurate masses of all toxins found from all venoms analysed can this way be retrieved from the LC-MS data and correlated to toxin IDs retrieved from the HT venomics data. The current main drawback in this analytical workflow is the tedious manual integration of all peak areas found in each venom analysed. In one of our previous studies, a similar workflow was demonstrated with venoms of Australian elapids for which the LC-MS data was indeed labouriously integrated manually (and the HT venomics results were not included) [10]. This drawback of manually integrating the intensities of all toxin *m*/*z* values found for each venom (and in this endeavour, only considering the highest charge state per toxin) is illustrated by the many months it took to process all LC-MS data. The current study presents automated peak area integration and sorting of all LC-MS data to overcome this drawback. Additionally, it includes HT venomics to correlate toxin masses used for venom variation analysis with toxin IDs.

We have validated this approach using 31 venoms of brown snakes (*Pseudonaja* spp.) and taipans (*Oxyuranus* spp.), two genera that form together the monophyletic *Pseudonaja*-*Oxyuranus* clade. This clade is well known for having some of the most lethal venoms of any snake species in the world [23] and has clinical relevance on the Australasian continent [24,25]. Our dataset covers all twelve documented species of this monophyletic clade, including several widespread localities. The LC-MS dataset was taken from our previous study, where the LC-MS data were manually integrated and processed [10]. This was carried out to validate the automatic data extraction procedure presented in the current study with the previous manual peak area integration [10]. By demonstrating and validating the automated LC-MS data extraction process, we aim for high-throughput LC-MS data processing, thereby allowing us to rapidly investigate many venoms for their variation in the toxins present in each venom under study. The automatic data processing methodology presented here reduces data extraction and processing time by >99%.

## 2. Results and Discussion

In this study, we present a method that can automatically extract venom protein features from LC-MS data. These features are arrays of several properties of each toxin: peak retention time, peak area, peak width at half height, most abundant *m*/*z* value, and the deconvoluted mass. “Features” is a term utilized to avoid referring to toxins and their characteristics before the process that validates all the mentioned properties. We ratified our results by comparing them to the standard—manual—manner of data processing [26]. To achieve this, we compared the results of automatic data extraction and processing with those of our previous manually curated data of the same dataset [10]. Furthermore, to correlate accurate toxin masses with toxin IDs, proteomic analysis was applied to two pooled venom samples using our HT venomics approach [22]. We analysed one pool containing all *Pseudonaja* venom samples and one containing all *Oxyuranus* samples. The pooled proteomic data allowed us to study venom variability at the toxin-ID level for a set of toxins, instead of the previously published toxin-accurate mass level alone [10]. An overview of the pipeline presented in this study is shown in Figure 1.

### 2.1. Automatic Feature Extraction

First, the features, which include peak area, peak retention time, peak width, and deconvoluted mass for each peak considered to be a toxin, were extracted from the LC-MS data. The extraction was automatically performed by recognising the retention time window for each toxin—also known as dissection—and deconvoluting the different charge states of each protein toxin from the raw Bruker LC-MS data. This was carried out using the built-in functions of the DataAnalysis Bruker software (i.e., dissection and deconvolution) using standard parameters specified in Section 4.2. By following this procedure, the software is able to recognise a coherence between *m*/*z*-value peaks belonging to the same toxin (i.e., the different charge states measured by MS for each toxin) through the raw LC-MS data and combines their intensities under the same feature (i.e., all information extracted for each toxin found in each LC-MS analysis). Subsequently, the mass that corresponds to each toxin is automatically calculated from its features. The peak area for each toxin is given as the sum of all the intensities of all *m*/*z* values of all charge states of the same toxin. In total, from all venom samples, 1035 features were extracted by using this method. An example of three arbitrarily different features for one of the samples can be found in Table 1.

Next, the automatically extracted features for each venom were compared with the manually processed results obtained by van Thiel et al. [10], who used manual deconvolution and peak-by-peak integration of the same LC-MS data. Therefore, these manually extracted and processed data provide an excellent opportunity to compare manual and automated data processing of the same dataset. Using the manual approach, a total of 185 unique masses were included in the processed dataset. This resulted in the extraction of 485 unique peak height values among all 31 venoms analysed. These peak height values were obtained from the highest peak height of the most abundant *m*/*z* value of each protein. Thus, these peak height values represent the unique toxins found in each venom, developing the concept of a “toxin sample pair”. A toxin sample pair is the feature corresponding to a specific toxin in a specific venom sample. For this study, there is a need to make a differentiation between toxin sample pairs and features, as each feature found through the two different data processing methods (i.e., manual and automatic) is compared to validate the data processing methods. Using the automated method, we identified 1035 peak areas, one for each toxin sample pair. This implied that the automatic method was able to detect more than twice the number of peaks that were recognised manually (i.e., 1035 vs. 485). As previously mentioned, the peak area of each feature was the chosen parameter to study because it is the most reproducible parameter that directly relates to the relative amount of a toxin found in each venom over a large concentration span [27]—which was the case in our sample set. The areas of the automatically extracted peaks were compared to the peak heights of the manually extracted data (because the area was not included in the manually extracted data).

### 2.2. Comparison between Manually and Automatically Extracted Data Using the In-House Script

The comparison between the manual method and the automated method described in this research is between toxins coming from the same dataset in both scenarios. As the method described in this study identifies more unique toxins, only the manually extracted toxin sample pairs were compared to the automatically extracted ones to avoid bias. The toxin sample pairs extracted both manually and automatically were compared using three diverse approaches. The first approach accounts for both mass and most abundant *m*/*z*-value match with a certain error window. The second approach considers either the mass or the most abundant *m*/*z* value to perform the matching. However, either due to human or algorithmic deconvolution errors, there is the possibility that the corresponding mass or most abundant *m*/*z* value will not match between both datasets; therefore, to address this, our third approach considered neither the mass nor the most abundant *m*/*z* value to match. Instead, we inferred the same toxin due to the tight relationship between these two values (e.g., if we find a manually deconvoluted toxin sample pair has a mass of 1000 Da and a most abundant *m*/*z* value of 500 and, for the same venom, we find an automatically deconvoluted mass of 2000 Da and a most abundant *m*/*z* value of 1000, it can be inferred that they are the same toxin). These approaches consider both the most abundant *m*/*z* values and the accurate masses of a toxin, and similar *m*/*z* values and accurate masses match among all venom samples. These three approaches were chosen because they cover all corrections between manually deconvoluted and automatically deconvoluted spectra.

All three approaches considered error windows for all comparisons that ranged from 0 to ±2.5 Da—all of which are explained hereafter. However, the three approaches vary in how close these two values between the toxins must be for them to be recognised as the same toxin, because if the mass and *m*/*z*-value match between deconvolution methods, we are more confident of them being the same protein, and thus can make the error window larger. Thus, for the first approach, both the *m*/*z* value and the accurate mass of two different toxins must be quite similar (the thresholds used were ±2 Da and ±1.1 *m*/*z* value) for them to be recognised as the same one. For the second approach, the procedure of the first one is followed, and the rest of the toxins are checked for similarities (the thresholds used were ±2 Da or ±1.1 *m*/*z* value) and recognised as the same. Lastly, for the third method, the procedure of the second approach was followed, and the remaining toxins were further compared. It was checked whether these yet-unmatched toxins contained accurate masses which, when divided by the mass of the suspected related toxin, the remainder of the operation was one of the following numbers: 0, 0.25, 0.333, 0.5, 0.75, or 0.666 (e.g., a protein with a mass of 1500 Da in the manually extracted dataset and 3000 Da in the automatically extracted dataset for the same venom would be considered the same). We selected these values to anticipate potential mass changes resulting from alterations in recognized charge states. They represent common relationships between charge values, allowing for full coverage: ratio of 2-1 (0.5), 3-1 (0.333), 1-4 (0.25), 2-3 (0.66), and 3-4 (0.75). By including these relationships, we look into variations in charge states and their relationship with the masses. This is carried out because manual error might lead to incorrect charge state or mass deconvolution. This approach makes it safe to assume that applying the third approach is equally valid to applying the first one; because we are analysing the same toxins, an approximately linear relationship between peak heights and areas should be found, as they both linearly increase with the concentration of the analyte. Thus, the relationship between the areas and their respective peak heights should remain constant if we are analysing similar toxins. The added benefit of the third method allows for more toxins to be recognised as matching with those of the manual extraction, as it includes all toxins from the first and second approaches, while adding the matching toxins that can be found as multiples of their mass values. A summary of this procedure can be found in Figure 2. A more detailed description of this procedure can be found in the Appendix A (Section: detailed comparison between manually and automatically extracted data).

Still, some of the toxins were not recognised as being the same because both their *m*/*z* values and masses were seen as different, while they did belong to the same toxin. Manually inspecting these toxins revealed that the difference was because the accurate mass had a proportional value to that found in the other dataset. The main reason this can happen is due to human error during the determination of the charge state. This issue can be circumvented by recognising all toxins in the automatically extracted dataset that have masses which division leaves one of the following numbers as a remainder: 0, 0.25, 0.333, 0.5, 0.75, or 0.666 with very narrow thread windows (≤0.005), and they are thus considered to be the same (e.g., a protein with mass 1500 Da in the manually extracted dataset and 3000 Da in the automatically extracted dataset for the same venom would be considered the same after the two previous comparison methods had not found any match). By applying this third method, 98.8% of the protein–venom pairs were correctly recognised, and 98.9% of the unique toxins were also recognised. The percentage of underestimated peak heights was 4.4% in this case, and 0.4% for the overestimated peak areas.

### 2.3. Comparison between Results of Manual and Automatic Approaches

The next step is to determine the compositional variability of venoms between the manual and automatic approaches. Therefore, we compared the principal component analysis (PCA) results between the two methods. The manual PCAs were taken from van Thiel et al. [10], and the automatic data processing procedure PCAs were made under the same settings. This allows for a direct comparison between both approaches. This process is included in the algorithms for extracting manual and automatic data that can be found in the Appendix A (ExtractionAutomatedData.jl and ExtractionManualData.jl).

When analysing all 31 venoms using 2 PCA dimensions, similar trends emerge between manual and automated approaches (Figure 3). Most notably, both PCA plots show a consistent pattern where *P*. *modesta* is completely distinct from the other venoms. However, both families are clearly differentiated as certain exact masses and are genus-specific. It is relevant that not only the scores (i.e., projection of the sample onto the principal components) should look similar, but also the loadings should hold similar weight for each of the toxins in each PC when comparing the manually extracted. This is because the distribution of the scores can be achieved by chance, but if the relevance of each variance is also the same, it means the data are comparable. Thus, the loadings of the first two PCs are plotted against the same values for the manually extracted data found in Figure 4. The PCs and scores corresponding to the *Oxyuranus* and *Pseudonaja* separately can be found in the Appendix A (Section: PCs and loadings of independent genera).

The general weight of all the variables is maintained constant throughout the different extraction methods. In some variables, the relevance of each feature (how much their values differ from 0) is not exactly the same. This is due to two main reasons that previously have been mentioned: (i) the use of areas instead of intensities in the automatic method, and (ii) the small percentage (5%) of toxin sample pairs that are either over or underestimated, as discussed in Section 2.2. There is an intrinsic importance to looking into the loadings of the PCAs, as the distributions found in Figure 3 might have been similar, but with no real similarities between the variables it used to generate that distribution. By looking at Figure 4, we can clearly see that relationship, and we can see how the importance given to each individual value is highly similar between manually and automatically extracted data. However, the key difference between both methods is that by using the automatic approach, the information was obtained faster, more robustly, and more accurately. Manual analysis is labour-intensive and time-consuming, whereas our novel automated approach provides a high-throughput platform that opens new possibilities in studying venom variation, making it achievable to evaluate large venom datasets rapidly and robustly. To investigate venom variability at the toxin level (i.e., toxin IDs), instead of only looking at their accurate masses, their accurate masses need to be correlated to their toxin IDs, which can be completed with the help of proteomics analyses. To demonstrate this procedure, we performed our recently established HT venomics [22] on pooled venom samples of the brown snakes (*Pseudonaja* spp.) and of the taipans (*Oxyuranus* spp.). Then, we used these data to correlate toxin IDs to their accurate masses for further investigating the data. The relevance of looking into toxin IDs is paramount for being able to correlate toxin masses to their toxin IDs. Knowing the toxin IDs of the toxins allows us to study toxin variation at the toxin ID level, thereby knowing which actual toxin is being studied. In addition, when looking into intact masses of toxins, their potential Post-translational Modifications (PTMs) can be investigated. When looking into these intact masses, possible variations in mass compared to exact masses given by Mascot (based on their amino acid sequence alone) due to PTMs can be revealed. This is the case if there is a mass difference between an accurate mass of a toxin measured and the mass given by the Mascot derived from its amino acid sequence alone and pinpoints PTMs. PTMs can alter the bioactivity of toxins and as such are important for their toxicity and resulting pathologies. Some PTMs, such as glycosylations, can drastically increase the immunogenicity of proteins, which is important knowledge for antivenom production.

### 2.4. High-Throughput Venomics

To identify all toxin IDs, venomics was performed on two pooled venom samples consisting of either all *Pseudonaja* species or all *Oxyuranus* species. Bottom-up venomics revealed that a total of 50 toxins were found within both pooled samples (Figure 5). In the pooled *Oxyuranus* venoms, we identified five toxin superfamilies: three-finger toxin (3FTx), phospholipase A2 (PLA2), Kunitz-type serine protease inhibitor (KTSPI), natriuretic peptide (NP), and Cysteine-rich venom protein (CRVP). All identified toxins are common among taipan venoms [28,29,30]. Surprisingly, venom factor X (vFX) and venom factor V (vFV), which together make up the prothrombinase complex, were not detected (Figure 5). Furthermore, no subunits of the pre-synaptic “taipoxin” complex were detected, though this toxin is a characteristic of venoms from the *Oxyuranus* genus (Figure 5 and Figure 6) [31]. In the pooled *Pseudonaja* venoms, we identified seven toxin superfamilies: 3FTx, PLA2, KTSPI, NP, CRVP, vFX, and vFV (Figure 5). All identified toxins are common among brown snake venoms [30,32,33,34]. In contrast to the pooled taipan venom, we did identify both vFX and vFV in the pooled *Pseudonaja* venom sample, which hints at, on average, a higher concentration of these proteins in this pooled venom sample. A recent study generating the most comprehensive *Pseudonaja textilis* venom gland transcriptome to date revealed the presence of at least 18 toxin superfamilies, of which at least 9 were actually expressed in venom [35]. We identified 7 out of 9 families among all brown snakes, with only 2 minor venom components (i.e., SVMP and 5′-nucleotidase) being unidentified.

We were able to identify all major toxin superfamilies in *Oxyuranus* and *Pseudonaja* venoms. From this information, we matched the exact masses calculated from each venom sample to the exact masses of the toxin IDs from the venomics. These toxins are obtained by an in-house written script that can extract the amino acid sequence of each protein from a toxin database (UniProt) and sum the exact mass of each amino acid while also considering the disulphide bonds in each toxin. Afterwards, the most abundant mass is also derived from the isotopic information of each amino acid. The total number of toxins that we were able to match among both pooled venoms include four venom gene families: 3FTx, PLA2, KTSPI, and NP. This shows that we were able to identify the three most common venom protein families (i.e., 3FTx, PLA2, and KTSPI) among taipan and brown snake venoms [28,29,30,32,34]. Unfortunately, not all toxins could be matched to their exact mass values. The main issue is that toxins undergo post-translational modifications, such as glycosylation and amidation [36,37,38]). These modifications affect their exact mass and are therefore difficult to predict. Furthermore, higher-mass toxins may break down during the separation and thus are detected as lower masses than that of the intact protein complexes. Compositional variation among the presence of 3FTx, PLA2, and KTSPI among brown snakes and taipans was observed, and we previously showed this variation can be explained by a combination of the absence/presence and differential abundance of venom toxins [10]. Although pooled venoms do not accurately reflect the total makeup of venom in each genus, proteomics analysis of each individual venom was not possible due to instrument-time restrictions. However, pooling the venoms and studying their toxins-IDs does allow for comparing the same toxin in the two different genera. Furthermore, the proteomics toxin IDs retrieved were used in the first place to correlate toxin-accurate masses (used for studying toxin variation) to their toxin IDs. It should then be noted that this is a relative qualitative comparison.

While the measurements did not find all expected present toxins, we did observe several biologically interesting patterns. The PLA2 OS1 (UniProt protein ID: Q4VRI5) was identified in all *O. scutellatus* populations, except for the isolated Northern Territory individual that did not possess this toxin. This might suggest that this population lost this PLA2 isoform, or might be an artefact of sample selection. The highly neurotoxic PLA2 OS2 (UniProt protein ID: Q45Z47) was only observed in *O. scutellatus* male individuals within the Cooktown and Saibai Island populations, whereas it was absent in both males of the Gladstone population. A short-chain neurotoxin (UniProt protein ID: A7X4S0) was identified in most taipans, except for *O. temporalis* venoms and the *O*. *microleptidus* Coober Pedy individual. Textilinin-3 (UniProt protein ID: Q90W99) was only identified in *P. textilis* males (n = 3) while being absent in the only female. This could potentially be an example of venom variation related to sex. Furthermore, our results also suggest that a short-chain neurotoxin isoform (UniProt protein ID: Q9W7K0) might be a clade-specific toxin, as it was only identified in *P. inframacula, P. aspidorhyncha*, and *P. affinis* venoms. While we must be cautious in overinterpreting any biological patterns without a representative sample size, our new approach allows for the rapid processing and analysis of large venom datasets.

## 3. Conclusions

Snake venoms are complex mixtures consisting of multiple toxins, and their compositional variation can be analysed by LC-MS. As most of the snake venom toxins are proteins, this makes it difficult for fast-paced analysis and data processing of many venom samples. Our study provides a new approach allowing for fast and accurate data extraction and analysis of numerous LC-MS data. This drastically reduces the amount of time spent on the tedious manual data processing of the analyses. We have demonstrated that our automatic data processing approach significantly outperforms manual data analysis for LC-MS data measurements of snake venoms. This provided a comprehensive dataset of the presence of venom toxins (i.e., their accurate masses) together with their relative intensities (extracted from the MS data) and allowed us to visualize both interspecific as well as intraspecific venom variation in a monophyletic clade of clinically important Australian elapids. Studying venoms with the here presented approach can help in better understanding venom composition and venom variation within venomous species. Also, together with the proteomics data, it can be used to obtain a better idea of possible PTMs present on venoms, which is important knowledge for antivenom production, as some PTMs (such as glycosylations) can influence the immunogenicity of toxins. Investigating this versatility in possible PTMs on venom toxins is conducted in practice by calculating mass differences between exact masses given by Mascot, which are calculated from toxin amino acid sequences alone, and accurate masses found by LC-MS. Knowing which PTMs are on toxins is, for some toxins, of paramount importance due to their involvement in the toxin’s bioactivity and, as such, its possible contribution to pathology. This is especially true when the PTM is a glycosylation, because it is able to modify the toxin’s immunogenicity. LC-MS and proteomics analysis of venoms with a focus on PTMs is, however, out of scope for this research but is the focus of one of our currently ongoing studies. The information that can be measured with the here presented research can be translated into valuable knowledge for clinical practice. Although demonstrated in this study for Australian elapids, with a focus on validation-type research, this methodology can straightforwardly be extrapolated to other venomous species and the toxins in their venoms.

## 4. Materials and Methods

The Liquid Chromatography–Mass Spectrometry (LC-MS) data used in this study was the same dataset used in the study by van Thiel et al. [10]. Thus, the LC-MS part of the workflow here described is given by van Thiel et al. [10]. In this study, additional HT venomics has been measured according to Slagboom et al. [22], for the correlation of toxin-accurate masses to toxin IDs.

### 4.1. Manual Processing of LC-MS Venom Variation Data

Bruker Compass software was used for the LC-MS-data analysis. The LC-MS data were acquired from van Thiel et al. [10] and it was manually analysed as follows: first, the average mass spectrum was extracted from the total MS-chromatogram using the Total Ion Current (TIC). This was carried out from a retention time of 0 min to 50 min. Subsequently, deconvolution of the charge states from each relevant toxin in the average mass spectrum was performed to yield the toxin-accurate masses, resulting in the monoisotopic masses of the toxins. Then, the Extracted Ion Currents (EICs) of the highest charge state of each toxin were plotted for the 10–15 highest abundant toxins. By plotting these highest-intensity charge states resulting in the corresponding EICs, the peak area integration value of the toxin for each EIC was then retrieved. For each of the EICs, their monoisotopic mass, *m*/*z* -value, and corresponding highest peak height were manually extracted for further analysis. Then, to confirm either the presence or absence of each mass among all other tested venom samples, all raw MS data were manually screened for the charge states found for each of the abundant masses. These masses were considered the same if they were all within 1 Da range from each other and had a similar retention time. This resulted in two comparative datasets: one including all taipan (*Oxyuranus* spp.) venoms, and one including all brown snake (*Pseudonaja* spp.) venoms.

### 4.2. Automated Data Extraction, Cleaning and Analysis of LC-MS Venom Variation Data

Dissection, a process by which compounds are found and their *m*/*z* values are grouped under the same unique mass, and deconvolution of the raw LC-MS data was performed for each individual venom analysed by Bruker DataAnalysis software to extract all features from the LC-MS analyses. These features consisted of all dissected potential toxins found in the MS data and included their corresponding properties, such as peak retention time, peak area, most abundant *m*/*z* value, and deconvoluted mass. Thus, samples are defined as all features belonging to an analysed venom. To repeat for clarity, each feature found in each venom contains the following information on a toxin: peak retention time, peak area, most abundant *m*/*z* value, and deconvoluted toxin-accurate mass.

The parameters for dissection were as follows: algorithm, version 3.0 (MS); sensitivity, 99%; area threshold, off; absolute intensity threshold, 1000; min. peak valley, 10%; internal S/N threshold, 3; max. number of overlapping compounds, 15; cut-off intensity, 0.01%; and recalculated precursor mass was ticked. The parameters for deconvolution were as follows: for peptides/small molecules, adduct ions, +H; deconvolute, MS; abundance cutoff [%], 1; and maximum charge, auto. Proteomics CHNO, excluding reporter ions and, creating a neutral spectrum were ticked. The output of this extraction procedure was a folder containing one .CSV file per venom, which included a list of the extracted toxin features.

Next, the toxin features within each sample were filtered by checking whether the same feature was found more than once within the same sample. Two toxin features were defined as being the same when there was a difference of, maximally, 1.5 *m*/*z* value or 2 Da in mass, and they eluted at the same retention time (defined as the top of the peak within a time frame of 3 times the standard deviation of the compound peak). These repeated toxin features were filtered out by generating a new feature with the added areas of both. This was carried out to consider an error found by us in the software dissection tool, where the same features were included several times within one sample due to the software creating two different features for the same toxin, either by recognising they had a different charge state or a different most common *m*/*z* value. The area of the remaining feature was the sum of itself plus the deleted one. This way, a list is obtained for each sample, containing all the features within the sample. Afterwards, all features within each sample that were previously (manually) processed by van Thiel et al. were used for data comparison between both studies to test the validity of the automated extraction method [10]. This resulted in a list that contained all matching features per sample, with the toxin abundances per accurate mass found by van Thiel and colleagues [10]. This list can be found in the Appendix A (Matrix of Toxins.xlsx). After checking the validity of the comparison between both datasets, as described and discussed in Section 2, the same dimensionality reductions by PCA were applied, and compared to van Thiel et al. [10].

### 4.3. High-Throughput Venomics

HT proteomics materials and methods can be found in the Appendix A (Section: HT venomics materials and methods), which was also detailed in [22].

## Figures and Tables

**Figure 1 toxins-16-00181-f001:**
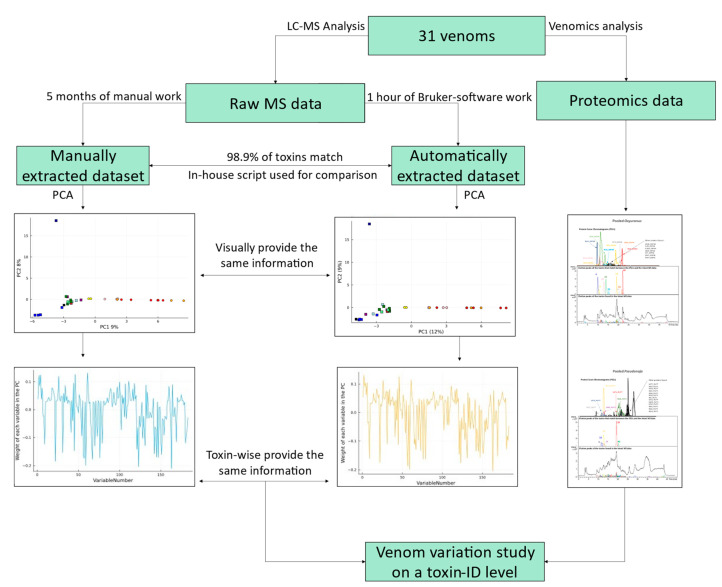
Summary of the analytical methodology used to investigate venom variation in the monophyletic *Pseudonaja*–*Oxyuranus* clade. The analytical workflow starts with LC-MS analysis of the 31 venoms included in this study, with an additional high-throughput (HT) venomics analysis of pooled venoms from *Pseudonaja* spp. and *Oxyuranus* spp. The LC-MS data were analysed both manually and automatically, both providing the same information. This is finally compared with the HT venomics data to analyse venom variation at the toxin-ID level.

**Figure 2 toxins-16-00181-f002:**
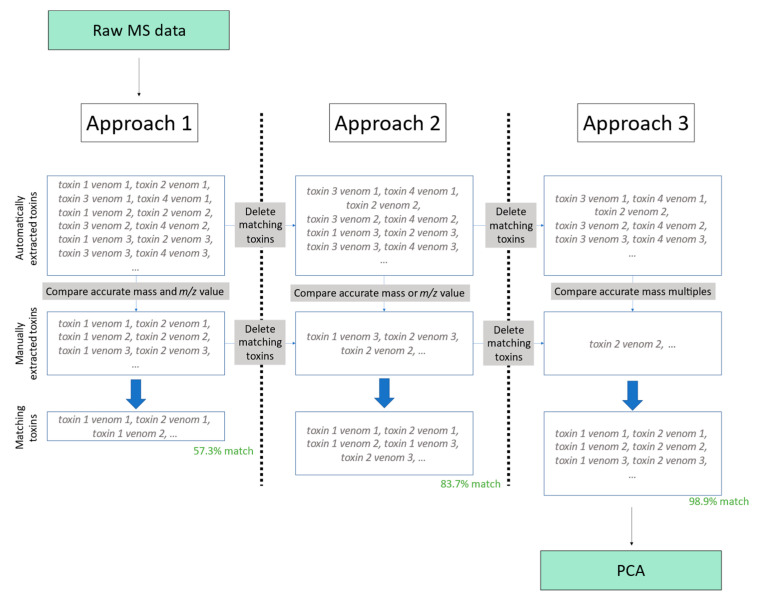
Summary of the three approaches taken for the comparison between the two methods. The data processing workflows start with the data obtained from the Raw MS data. The toxin sample pairs are compared by the three different methods. A matching of 98.9% of the toxin sample pairs was obtained by the best processing workflow (i.e., approach 3) on which finally PCA was performed.

**Figure 3 toxins-16-00181-f003:**
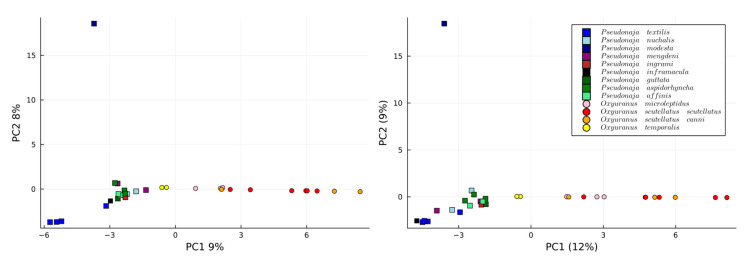
PCA representation of the manually extracted data (**left**) and the automatically extracted data (**right**) for the whole dataset. PC1 can differentiate between the two genera (*Pseudonaja*; squares) and (*Oxyuranus*; circles), whereas the PC2 distinguishes between the *P*. *modesta* and the other *Pseudonaja* species.

**Figure 4 toxins-16-00181-f004:**
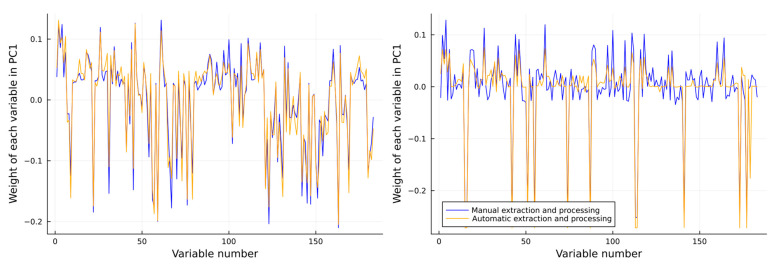
Representation of the loadings of the PCAs 1 and 2 of the manually extracted data (blue) and automatically extracted data (orange) for the whole dataset PCA. In both cases, the variables with the highest weights follow the same trend.

**Figure 5 toxins-16-00181-f005:**
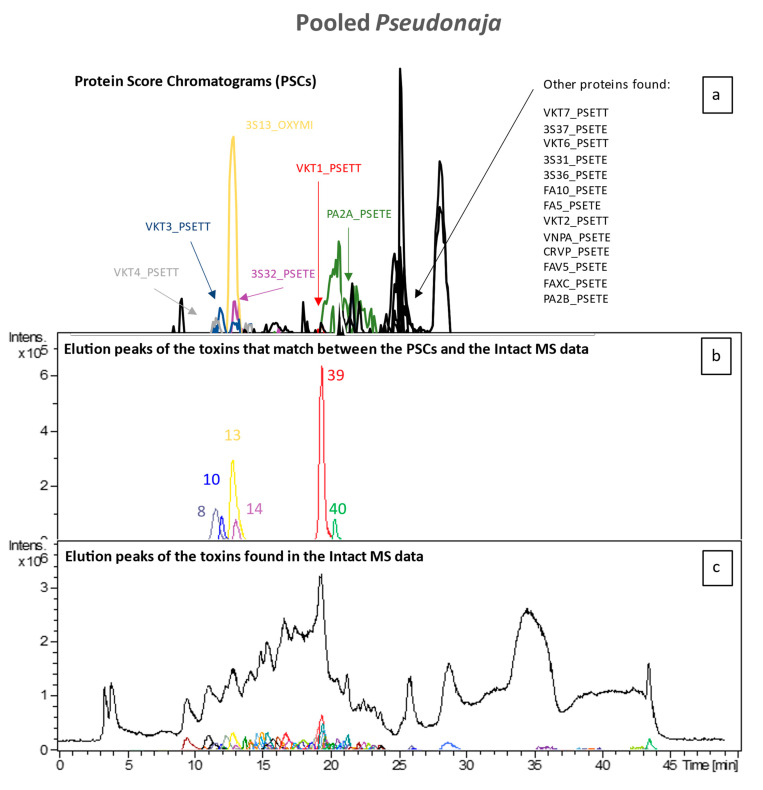
Merged results from the analysis of the pooled *Pseudonaja* venoms (**a**) and intact MS results (**b**,**c**). While the TIC and the chromatographic peaks of all toxins can be found in (**c**), (**b**) contains all chromatographic peaks of the toxins of which their accurate mass peaks matched with those of the PCSs in terms of retention time and peak shape. The coloured toxins in (**a**) represent these toxins, whereas the black ones represent those toxins for which accurate mass data could not be matched.

**Figure 6 toxins-16-00181-f006:**
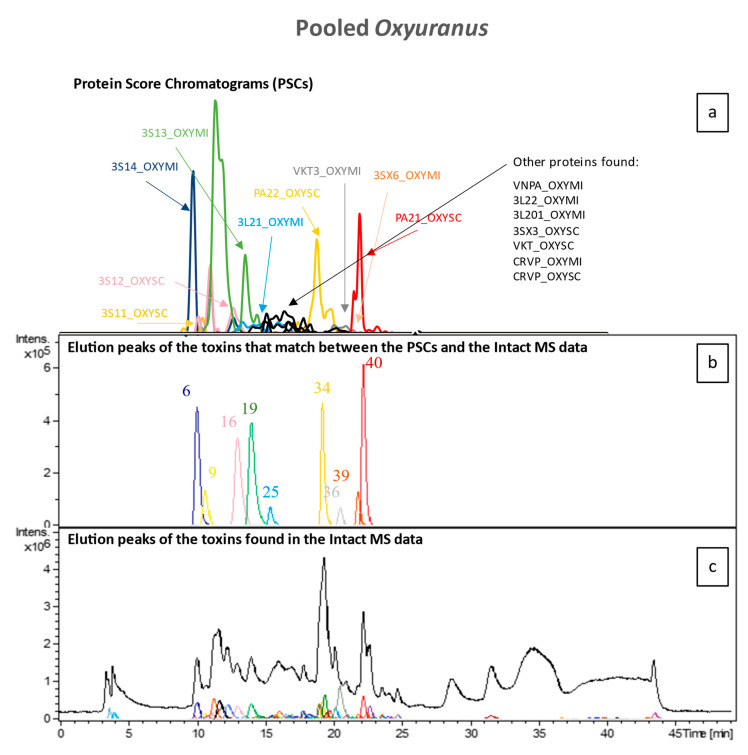
Merged results from the analysis of the pooled *Oxyuranus* venoms (**a**) and intact MS results (**b**,**c**). While the TIC and the chromatographic peaks of all toxins can be found in (**c**), (**b**) contains all chromatographic peaks of the toxins of which their accurate mass peaks matched with those of the PCSs in terms of retention time and peak shape. The coloured toxins in (**a**) represent these toxins, whereas the black ones represent those toxins for which accurate mass data could not be matched.

**Table 1 toxins-16-00181-t001:** Three arbitrarily chosen features in one sample are shown as an example.

# Peak	RT (min)	Area (Count·min)	FWHM (min)	MW (Da)	I (Count)	Max. *m*/*z* Value
57	24.7	233,095	0.3	15,149.29	13,716	3029.47
58	24.8	504,943	0.3	29,802.46	26,676	3313.84
59	25.1	343,360	0.3	14,948.22	18,931	2992.06

RT = Retention Time; FWHM = Full Width at Half Maximum; MW = Molecular Weight; I = Intensity.

## Data Availability

Data are contained within the article.

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
