# Peer review of "Studying Venom Toxin Variation Using Accurate Masses from Liquid Chromatography–Mass Spectrometry Coupled with Bioinformatic Tools"

_toxins, 2024, doi:10.3390/toxins16040181_

Round 1

Reviewer 1 Report

Comments and Suggestions for Authors

This manuscript presents a novel way for quickly analyzing several venom samples using automated procedures. The method utilizes LC-MS (Liquid Chromatography – Mass Spectrometry) to separate venom and analyze toxins accurately, in addition to freshly created bioinformatic scripts for efficient results. This approach enhances venom analysis by enabling quick and comprehensive investigations of venom diversity. The results propose using animal venoms as model systems for evolution and studying variations to better the development of antivenoms. The method proposed here is timely and I would recommend the publication of this work after addressing some few concerns which are as follows;

1.     Elaborate in the text to what extent is the presented pipeline scalable and applicable to other venomous species or toxin types? Are there considerations or modifications needed when applying this method to different datasets?

2.     In studying venom variability at the toxin-ID level for a set of toxins, what insights were gained that were not apparent when focusing solely on accurate toxin masses? How does this approach enhance our understanding of venom composition?

3.     Regarding the third approach, which involves matching toxins based on mass multiples, can you clarify how the decision was made to choose specific remainder values (0, 0.25, 0.333, 0.5, 0.75, or 0.666)? Were these values determined through empirical observations or theoretical considerations?

4.     In result section, ‘High Throughput Venomics’, can you clarify the rationale for choosing pooled venom samples for venomics analysis, specifically emphasising all Pseudonaja and all Oxyuranus species? To what extent do these pooled samples accurately reflect the total makeup of venom in each genus?

5.     Figure 4 illustrates the loadings of PCAs 1 and 2 for manually extracted data in blue and automatically extracted data in orange. Please elaborate on the importance of the variables with the highest weights and their effect on the entire evaluation.

Reviewer 2 Report

Comments and Suggestions for Authors

I consider that this work is relevant for the Special issue to which it was submitted.

I also consider that it is an interesting contribution to its specific field.

It is well written; I’ve made a few corrections, but I have asked questions and made suggestions with the aim of avoiding confusion about 2 terms used by the authors, and that I consider to be key in their work (please see attached file).

Comments on the Quality of English Language

It is well written; I’ve made a few corrections, but I have asked questions and made suggestions with the aim of avoiding confusion about 2 terms used by the authors, and that I consider to be key in their work (please see attached file).

Author Response

We thank the reviewer for all the changes they pointed out.  We addressed all comments from this reviewer, who made their comments in a pdf document of our manuscript. All small errors and word changes suggested by the reviewer we adjusted in the manuscript. In addition, the reviewer had a specific concern regarding two terms which use were not absolutely clear and that were paramount to the understanding of the manuscript. These terms were “integration” and “features”, which were indeed not fully clear. Changes were made throughout the manuscript to precisely use them, and we added extra descriptions about them in relevant lines such as [83, 127-128]. The included texts contain: “at the quantification of the peak areas -a.k.a., integration- and data processing stages” and “ ‘Features’ is a term utilised to avoid referring to toxins and their characteristics before the process that validates all the mentioned properties”.

Reviewer 3 Report

Comments and Suggestions for Authors

The authors performed a study about venom toxin variation using accurate masses from liquid chromatography-mass spectrometry coupled with bioinformatic tools. Although the study is of interest, some changes are needed:

1.     Please add some sentences focusing on how your research can be used in clinical practice

2.     Explain better the aim of your study, specifically on page page 2 lines 93-94  you wrote “…This was done to validate the automatic data extraction procedure presented in the current study with the previous manual integration”. Therefore give a more detailed reason, for which you have performed this study.

3.     Please explain better why you used the pool from Oxyuranus and Pseudonaja

4.     Improve the resolution of the Figure 3

5.     Please in the conclusions explain better what is the translational (into clinical point of view) improvement of your research.

Round 2

Reviewer 3 Report

Comments and Suggestions for Authors

The authors performed the changes and the article can be accepted for publication.